# Vesicoscopic Cross-Trigonal Ureteral Reimplantation for Vesicoureteral Reflux: Intermediate Results

**DOI:** 10.3390/children9020298

**Published:** 2022-02-21

**Authors:** Christian Kruppa, Guido Fitze, Katrin Schuchardt

**Affiliations:** Department of Pediatric Surgery, Universitätsklinikum der TU Dresden, 01307 Dresden, Germany; guido.fitze@uniklinikum-dresden.de (G.F.); katrin.schuchardt@uniklinikum-dresden.de (K.S.)

**Keywords:** vesicoureteral reflux, Cohen, cross-trigonal ureteral reimplantation, endoscopy, children

## Abstract

For the treatment of vesicoureteral reflux, the introduction of vesicoscopic procedures offers new perspectives for improving patient comfort and quality. Our aim was to examine whether minimally invasive vesicoscopic cross-trigonal ureteral reimplantation (VCUR) would meet expectations. Between 2012 and 2021, 99 girls and 35 boys with high-grade vesicoureteral reflux (VUR) underwent VCUR. For two boys, we failed to establish the pneumovesicum, leading to conversion to open surgery. The mean age was 4.5 years, ranging from 10 months to 18 years. VCUR was successfully performed in 132 patients, including 75 patients with bilateral VUR and 12 children with double ureters with unilateral or bilateral VUR, corresponding to a total of 229 operated ureters. The mean time of operation was 151 min for all patients. There were no perioperative complications, with the exception of three cases of pneumoperitoneum without consequences. Postoperatively, we recognized three cases of acute hydronephrosis, two of them required transient drainage. Three patients developed extravasation of urine after the postoperative removal of the transurethral catheter, rapidly resolved by new drainage. In two patients, we combined VCUR with laparoscopic heminephrectomy and opposite laparoscopic nephrectomy, respectively. Overall, mean postoperative hospital stay was 4.2 days. We observed recurrent VUR in seven ureters, resulting in a success rate for VCUR of 96.9%. These results demonstrate the feasibility of VCUR and its potential to displace open surgery with high safety and wide applicability.

## 1. Introduction

Vesicoureteral reflux is the unphysiological reflux of urine from the bladder back into the ureter and possibly into the kidney. It represents the most common pediatric uropathy, with a prevalence of 0.4–1.8% [1]. In 10–40% of children with pyelonephritis, VUR can be detected [2]. The main goal of treatment is to prevent upper urinary tract infections that can lead to renal scarring, thus preserving kidney function. The possibility that subclinical VUR may also lead to renal damage adds to the complexity of the disease.

The therapeutic options depend on factors such as the patient’s age, renal function, degree of reflux, parental acceptance, and frequency of febrile urinary tract infections. Accordingly, the spectrum of possible therapeutic approaches is broad, ranging from watchful waiting to prophylactic antibiotic medication, to more or less invasive surgical interventions. In view of the high spontaneous resolution rate of VUR in young children, at least the first year of life is basically reserved for conservative therapy [3].

In recent decades, the endoscopic injection of bulking agents into the submucosa of the refluxing ureteral orifice has become an accepted alternative therapy. The success rate may depend on the technique used, with the hydrodistention implantation technique (HIT) of the subureteral transurethral injection (STING) procedure proving advantageous [4]. The effectiveness of endoscopic injection treatment is reported very differently in studies. Influencing factors are, for example, the technical implementation and the experience of the surgeon, as well as the specifics of the patient. There is a clear correlation between VUR resolution rate and VUR grade. In their study, Friedmacher et al. recorded a resolution rate of 69.5% for the first injection in 1287 refluxing units with grade 4 and 5 [5]. For lower grade VUR, higher success rates are achieved, with 71% for grade 3, 83% for grade 2, and 98% for grade 1 [6]. Because endoscopic injection therapy is quick and easy to perform, painless, and has a very low complication rate, it is preferred by many pediatric urologists as an alternative to long-term antibiotic prophylaxis for suitable patients with a low VUR grade and good renal function [7].

Nevertheless, the more invasive surgical approaches to VUR therapy retain their indication in high-grade VUR, particularly in combination with the functional impairment of the kidney. Success rates for open surgery range from 95% to 98% [8]. The surgical treatment of VUR has undergone remarkable development. Established open surgery procedures are now increasingly competing with their minimally invasive implementations. The aim of future surgical standards must be to prevent pyelonephritis safely and reproducibly, while causing little surgical trauma and minimal postoperative morbidity, and maintaining a high level of acceptance by the parents of children with VUR [9].

Persuasive urological results are confirmed for open surgery, but in contrast we see the fundamental disadvantage of a sufficient incision of the abdominal wall and, if necessary, the bladder for access. Various endoscopic modifications of established open procedures have already been described [10,11,12,13]. Since the introduction of the pneumovesicoscopic cross-trigonal ureteric reimplantation by Yeoung et al., 2005, VCUR has attracted attention [14]. This is reflected in an increasing number of publications reporting positive results with respect to the high requirements for safety, success rate, and patient comfort. Advantages of VCUR are expected due to its minimal invasiveness, which should result in shorter hospitalization times, less need for analgesics, and barely visible scars [15,16,17,18,19].

However, due to its technical challenges, VCUR is still far from being used regularly. The aim of our study was to find out whether vesicoscopic cross-trigonal ureteral reimplantation can be performed in everyday life and whether children can benefit from it.

## 2. Materials and Methods

In this retrospective study, we report on all patients who underwent VCUR in our institution from January 2012 to July 2021. All children had proven VUR with a history of recurrent chemoprophylaxis-resistant pyelonephritis or long-term persistent high-grade VUR or VUR and renal scarring. On at least one side, among the 132 children, a VUR grade 5 was conspicuous in 37 cases (28%), a grade 4 in 57 cases (43%), a grade 3 in 35 cases (27%), and a grade 2 in 3 cases (3%). Indications were determined according to the recommendations of the American Urological Association [20].

Complete medical reports, including voiding cystogram respectively contrast-enhanced voiding urosonography and 99mTC-MAG3 renal scintigraphy data were available for every patient. Informed consent was obtained from all subjects involved in the study.

Of the 132 patients, 75 had bilateral refluxive ureters, and 45 were unilateral. A total of 12 children had refluxive duplicate ureters on at least one side, including 8 patients with refluxive duplicate ureters on one side and a refluxive single ureter on the other side, and 1 girl with bilateral duplicated ureters with VUR. A total of 229 ureters were reimplanted. The mean age of the children was 4.4 years (range 10 months to 18 years). The operating time was measured from the beginning of the cystoscopy to skin closure. Our study reflects the experience of a single surgeon, yet parts of the operations were also performed by assistants for training reasons.

Our surgical approach follows, with modifications as previously described by Valla et al., the endoscopic variant of Cohen’s ureteral reimplantation [17,21]. Patients are positioned in modified lithotomy position with the thighs elevated and abducted. The procedure begins with cystoscopy, preferably with a 70° angled lens, and the bladder is completely filled, avoiding excessive filling pressure. Appropriate to the patient´s age, three small incisions are made a few centimeters above the symphysis for the midline 6 mm optic trocar and two 3.9 mm trocars laterally. Under cystoscopic guidance, large 2/0 transfixation sutures are passed percutaneously through the bladder cavity around the incisions. The trocars are inserted into the bladder while the corresponding retaining suture is under traction (Figure 1).

The cystoscope is removed, a 30° 5 mm telescope is introduced, and the bladder is emptied of saline, replacing it with CO_2_ under light pressure (7 mmHg). The camera is adjusted and attached to a piezoelectrically lockable holding device (Endocrane^®^, Storz SE & Co. KG, Tuttlingen, Germany) (Figure 2).

Now, the ureter is intubated with a 3 or 4 Fr. silicone tube, followed by subtle circumferential electrical mucosa incision with a fine dissecting needle (Figure 3). The ureteral release is continued using an electrical hook for at least 4 to 6 cm until it can be positioned without tension to the opposite ostium (Figure 4). Duplicate ureters are intubated twice and mobilized together. If bilateral, the submucosal tunnel is created between the two ostial incisions. If not, an additional mucosal incision is made 1 cm canially of the opposite regular ostium (Figure 5).

Submucosal tunnel preparation using curved scissors often requires some patience. When the tunnel is finished, the refluxive ureters are now shifted to the opposite side equivalent to the open approach. An ipsilateral suture fixes the ureter to the detrusor (Figure 6).

After sparingly shortening and spatulating, the neoostia are created with 5/0 polyglyconate or polyglactin sutures (Figure 7). Depending on the age of the patient, a submucosal ureteral course of 3 to 5 cm in length can thus be achieved. If the finding is unilateral, the original mucosa incision must be closed (Figure 8). Only in exceptional cases, e.g., renal insufficiency, are the ureters splinted for 6 days by percutaneously inserted splints. Finally, the bladder is drained through a balloon catheter for at least 1 to 2 days. After the trocars have been removed, the trocar incisions are only adapted cutaneously, using skin patch strips. The patients are discharged home with problem-free, spontaneous micturition, safe well-being, and regular ultrasound findings.

All patients are sonographed after 4 weeks, 6 months, and 12 months, and then remain under long-term pediatric nephrological control. Voiding cystourethrogram (VCUG) or contrast-enhanced voiding urosonography is performed on all children who have postoperative UTIs, repeated pathological urine findings, or sonographic abnormalities, such as growth disorders or hydronephrosis.

## 3. Results

VCUR was successfully performed on 132 patients. In two additional patients at the beginning of our series, it was not possible to establish the pneumovesicum. Trocardial dislocations led to extravesical leakage of saline. With the bladder lumen minimized as a result, the operations had to be continued openly.

Overall, we recorded an average total surgical time of 151 min (SD 42) from the beginning of the cystoscopy to the completion of the skin closure. The average operating time for the first 40 operations was 171 min (SD 45), and for the last 40 patients it was 128 min (SD 36). The procedures lasted the longest in the group with unilateral or bilateral double ureter (194 min average, 290 min max.). The mean operating time was 158 min (100 to 230 min.) for bilateral VUR, and 128 min (72 to 207 min) for unilateral VUR. Statistically, a significant connection was found between the reduction in operating times and the experience of the surgeon, in line with the learning curve, according to a Spearman rank correlation coefficient rs = −0.45, *p* < 0.001. Intraoperative blood loss was negligible.

The average postoperative length of stay of the children in the hospital until they were discharged home was 4.2 days (min. 1, max. 13 days); for the last 40 patients, 3.3 days. The mean follow-up time was 44 months (min. 6 months, max. 9.5 years).

We operated on two children with chronic renal insufficiency and bilateral VUR. Both had an unremarkable clinical course. In both cases, the ureters were relieved for 6 days by percutaneous splints.

Two children underwent laparoscopic kidney surgery in the same session. One child underwent a right nephrectomy in addition to the VCUR on the left, whereby the refluxive right megaureter was released distally by vesicoscopy. In addition, for the second child, the VCUR on one side was combined with a laparoscopic upper heminephrectomy on the opposite side. The further course was unremarkable for both children; they left the hospital on the 4th respectively 7th postoperative day.

In two patients, we noticed a pneumoperitoneum during the operation, which did not affect the operation, and which disappeared without consequences within a few hours after the procedure.

Three children had postoperative voiding problems and abdominal pain caused by extravasation of urine after Foley catheter removal. In all of them, the situation returned to normal after renewed continuous catheterization for 5 days.

Three children with grade 4 VUR bilateral had an obstruction with painful hydronephrosis postoperatively. In one of these patients, the condition improved spontaneously; one child received percutaneous nephrostomies for a week. The third (ureter fissus bilaterally) had a urinary tract infection; in this case, the ureters were openly surgically stented percutaneously transvesically. Two other children with postoperatively pathological urine and suspected UTIs were treated with antibiotics alone.

Postoperative VUR diagnostics were undertaken in 17 patients with clinical or sonographic abnormalities. A recurrent VUR was found for 7 ureters in 6 patients (5 girls with bilateral VUR, 1 boy with unilateral VUR). This corresponds to a success rate of 95.5% for the number of patients and 96.9% for the ureters. The re-operations showed that the affected ureters had slipped back into their original positions.

## 4. Discussion

VUR surgery is one of the most frequent urologic procedures in children. Historically, a variety of beneficial surgical techniques have been developed to prevent VUR by extending the submucosal course of the ureter. In order to make the necessary corrections, sufficient surgical accesses must be created for the open methods by definition. This results in appropriate intraoperative tissue trauma with blood loss and postoperative pain and, in the case of transvesical procedures, additional bladder irritation, hematuria, and the need for prolonged uncomfortable catheterizations [16,22].

For the operation of VUR, the different endoscopic procedures should be equal in terms of applicability. However, laparoscopic antirefluxplasty according to Lich-Gregoir turns an extraperitoneal procedure into a transperitoneal one [23]. Furthermore, bilateral surgery is still not recommended because temporary bladder voiding dysfunction may occur [24]. In their 2005 study, Soh et al. compared the results of different pneumovesicoscopic ureteral reimplantation. Twelve female patients were each operated on according to Cohen and Leadbetter-Politano. In terms of resolution of reflux, duration of catheterization, and length of hospital stay (mean 3.6 days), both procedures yielded the same result. The operating time of the patients treated according to Leadbetter-Politano was on average one hour longer than the mean time of the Cohen group. The additional step of retrovesical ureteral preparation in the Leadbetter-Politano procedure makes this procedure more complex and invasive [13,21,25].

The results presented here include experiences starting from the first patient. The learning curve, which is one for the entire team, should be similar to that of other endoscopic surgeries. Once the technical requirements and surgical technology are established, any reconstructive-experienced endoscopic surgeon will be able to perform VCUR successfully. The use of the piezoelectric holding arm relieves the assistant almost completely and makes VCUR a one-person operation (Figure 2).

In our series, the operation time improved from 174 min for the first 40 VCURs to 128 min for the last 40, matching the literature [19,26]. As expected, VCURs for ureter duplex are among the most time-consuming in the series, owing to the less clear anatomy during preparation and the more difficult reimplantation of very delicate ureters [27].

Two patients after subureteral injection could be operated on with the VCUR without any problems, but local scarring or ureteral wall injection are probably strictly limiting in this case [18].

Since the trocar incisions are not closed at the end of the VCUR, there is a risk of extravasation of urine into the extravesical spatium. This was evident in three of our children with Foley catheter removal immediately postoperatively or on the first post-op day. We then extended the minimum duration of Foley catheter use from 1 to 2 days (from Patient 68), after which extravasation did not recur.

The subject of the discussion is also the applicability of VCUR with regard to the minimum age of the patients. Minimal working space and difficulties in establishing the pneumovesicum were named as limiting factors [15,17,27]. In our review, 23 patients were children of less than 18 months of age with antibiotic–refractory recurrent severe pyelonephritis. Although there were 11 children with bilateral VUR and another 5 children with double ureters (4 included VUR on the opposite side), the young age had no influence on the feasibility or the operation time (mean 149 min.). A helpful factor could be found in the image stabilization provided by the holding device. It ensures a minimum of movement at the trocars, which protects them from dislocation.

Since not all patients underwent VCUG postoperatively, there is the possibility of an objectively lower VCU-resolving success rate than the 96.9% in 229 ureters. Nevertheless, we refrain from routine VCUG controls postoperatively, with a very low probability of success in children who have no anamnestic or nephrological abnormalities. In our opinion, the high VUR resolution rate, together with the non-maximum sensitivity of the VUR diagnostics, lead to a disproportionate number of unnecessarily examined children. In addition, there is often a lack of acceptance by parents and children in the case of an inconspicuous clinical course [17,19,28].

Even after short-term radiologic success, the findings could deteriorate; for the patient, the clinical benefit is crucial. In order to find a definition for the success of a VUR operation that improves the comparability between the studies, further research must be carried out in the future [29,30].

It was noticeable that in all children who underwent reoperation because of recurrent VUR, the affected ureters had slipped back out of the tunnel and into their original positions. With this in mind, we introduced ispilateral external ureteropexy from Patient 91 onwards. This reduces the tension on the neoostium that is always present despite mobilization. Since then, we have not recorded a new recurrence.

## 5. Conclusions

VCUR proved to be an effective, age-independent, widely applicable, and minimally invasive option. It is possible to combine VCUR with laparoscopic interventions. The longer operating time than with open surgery is within tolerable limits. Similar success rates, short catheter stays, short hospitalizations and excellent cosmetics recommend VCUR as an alternative to open surgery although further studies are required.

For the endoscopic surgeon, VCUR offers the opportunity to develop skills in a safe environment.

## Figures and Tables

**Figure 1 children-09-00298-f001:**
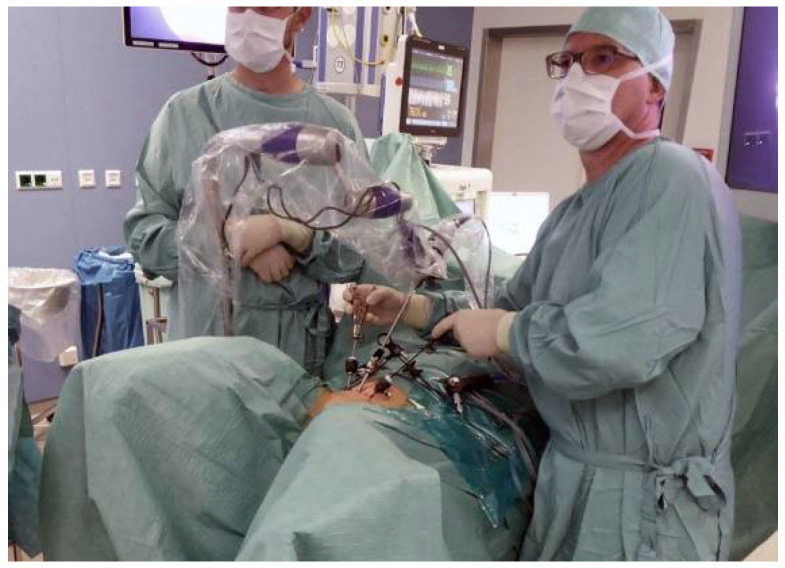
The patient is positioned in modified lithotomy position; the right-handed surgeon stands at the left; the camera is held by the holding device.

**Figure 2 children-09-00298-f002:**
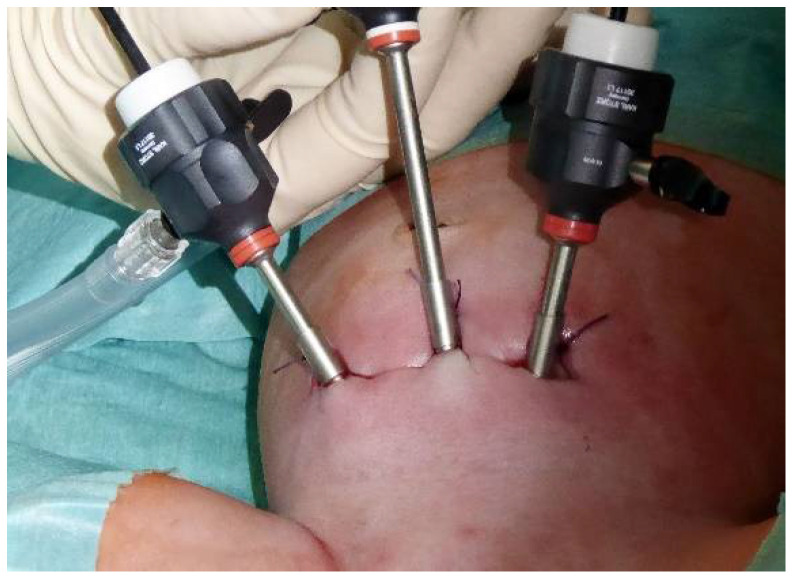
The trocars are placed under a cystoscopic view and attached to the transfixation sutures.

**Figure 3 children-09-00298-f003:**
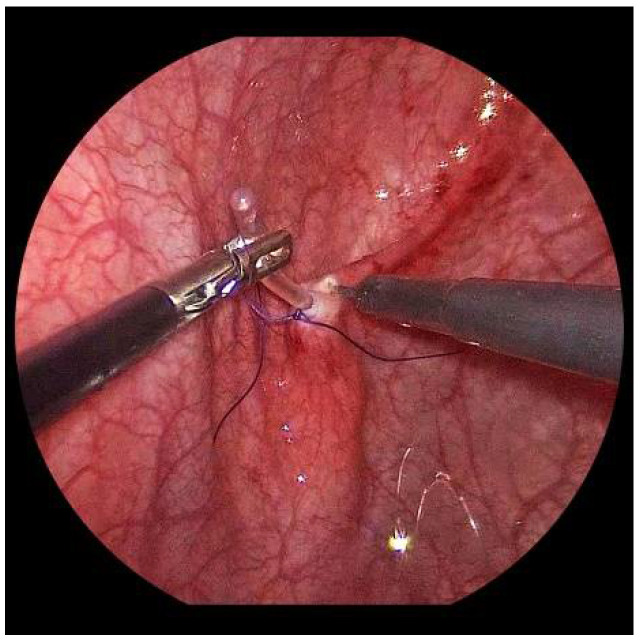
For a left-sided VCUR, the ostium is circumferentially incised with a fine needle while gently pulling on the intubated left ureter.

**Figure 4 children-09-00298-f004:**
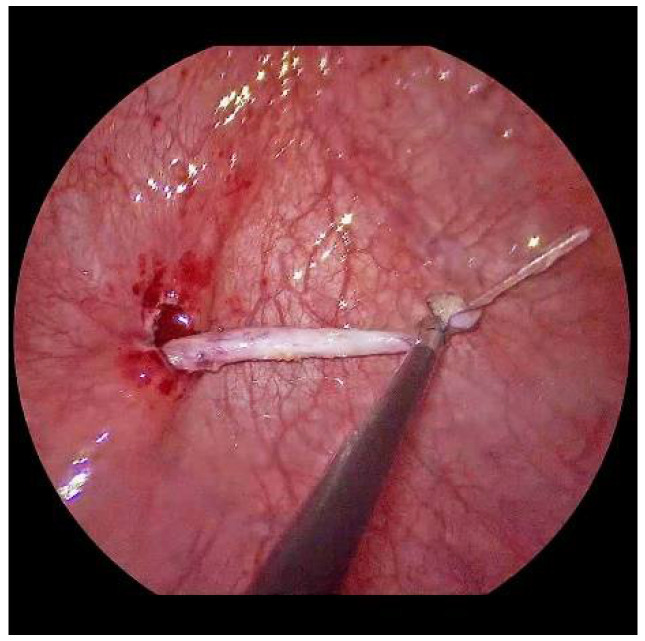
After mobilization, the left ureter can be shifted to the right ostium with little tension.

**Figure 5 children-09-00298-f005:**
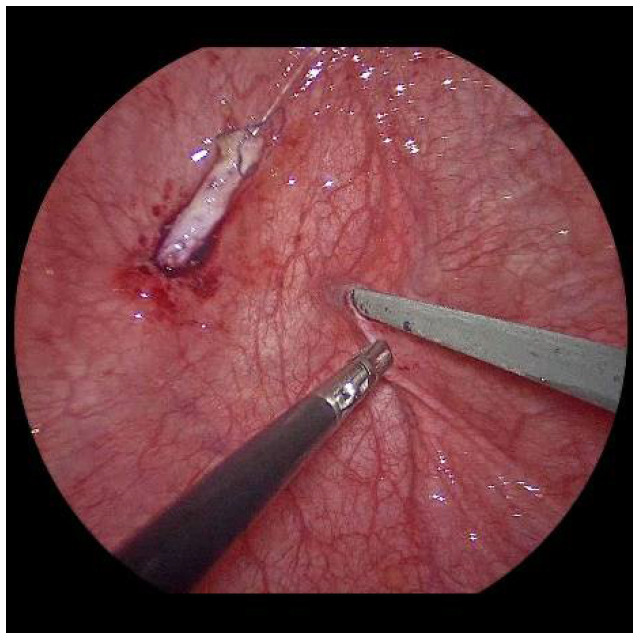
The submucosal tunnel is subtly prepared with curved scissors.

**Figure 6 children-09-00298-f006:**
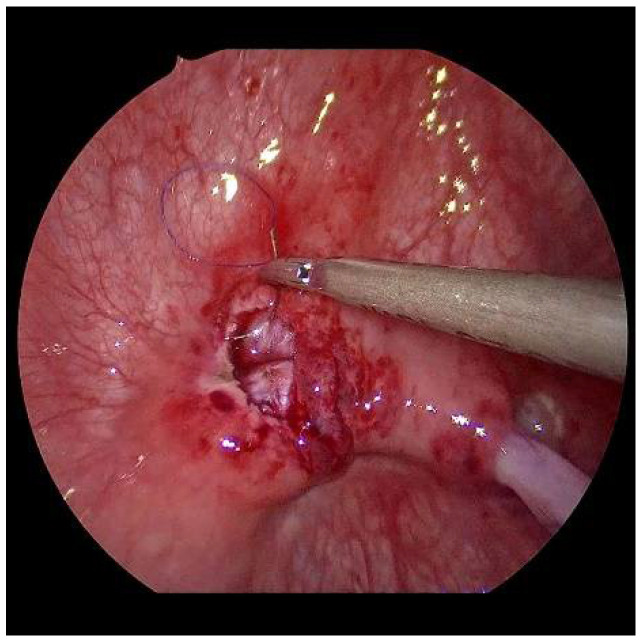
To reduce the tension on the neoostium, the ureter is pexed at the bladder entrance.

**Figure 7 children-09-00298-f007:**
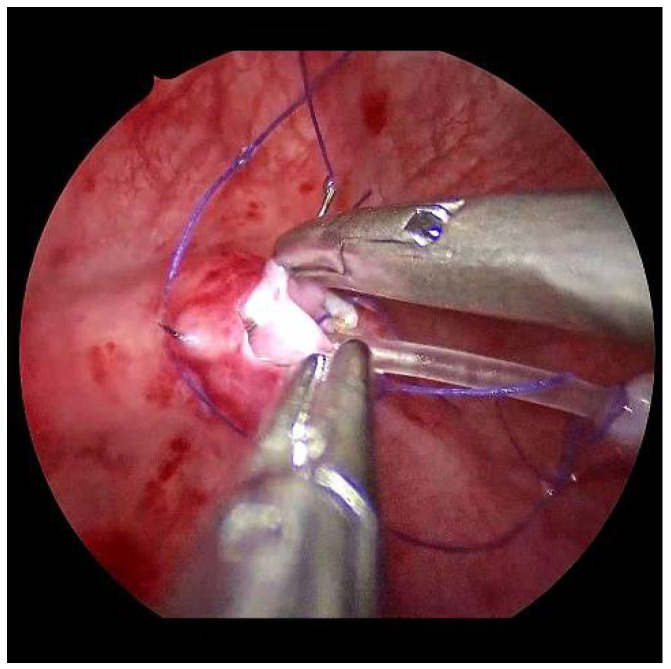
After sparingly shortening and spatulation, the neoostium is formed on the left.

**Figure 8 children-09-00298-f008:**
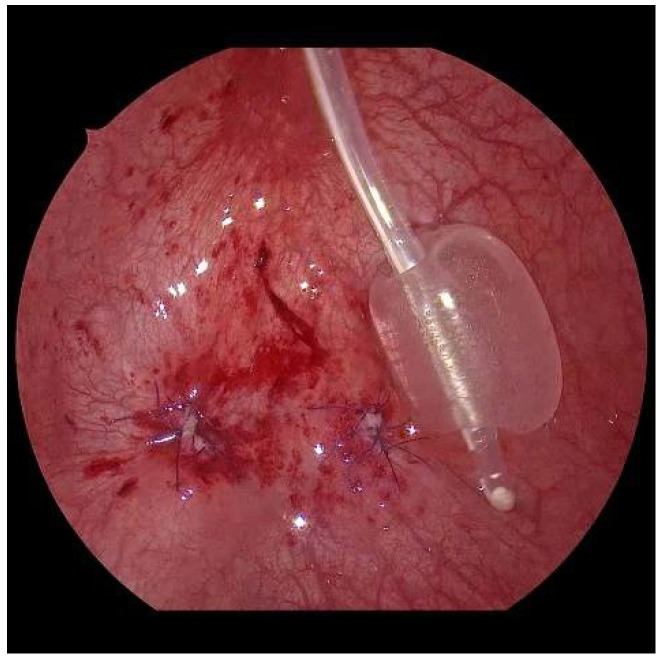
Aspect at the end, with closed mucosal defect on the right and the neoostium on the left.

## Data Availability

Not applicable.

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
