# Peer review of "Vesicoscopic Cross-Trigonal Ureteral Reimplantation for Vesicoureteral Reflux: Intermediate Results"

_children, 2022, doi:10.3390/children9020298_

Round 1

Reviewer 1 Report

Very interesting, novel and well written article without much to add from the reviewer side.

In the introduction it´s explained that endoscopic treatment has low long term success. This should be better described, percent of success varies between techniques and also the degree of pre-surgical reflux, the presence of UTD, previous injections and surgical experience are  determining factors in the results of endoscopic therapy according to the different studies published in this topic.

The article cited in introduction is from 2010, it should be better a newer systematic review, and also explained if the success rate are analyzing different grades of Reflux. It is well known that for VUR higher grades the success of endoscopic technique lowers, this is important as for a low grade reflux the success rate is higher that the general, and this leads us to choose lesser invasive surgery. Robotic, laparoscopic surgery even this elegant technique described in the article are invasive techniques, and should be reserved for high grade reflux as the authors claim is the case in their study.

Also, it is important to state exact grade of reflux, not just say all had high reflux.  x% grade IV, y% grade V.

Thank you very much for your manuscript

Author Response

Dear reviewer,

Thank you for your assessment and recommendations. I have expanded the introduction to the topic and tried to present the value of endoscopic therapy in more detail. 

For the classification of reflux grades, we have chosen the number of patients as the reference point. This better reflects the basic indication for surgery. 
If we were to use the number of ureters as a basis, this would result in slightly more low-grade VUR (due to findings on the opposite side or double units). 
In contrast to injection, we do not see a correlation between the degree of reflux and the success/recurrence rate with the cohen method.

With best regards

C. Kruppa

Reviewer 2 Report

Thank you for allowing me to review the manuscript entitled " Vesicoscopic cross trigonal ureteral reimplantation for vesicoureteral reflux: intermediate results. The authors describe a single center experience. It is not specified if this is a single surgeon experience. This information would be helpful.

They authors also describe a 4-5 cm mobilization of the ureter followed by submucosal tunnel creation and shortening and spatulation, followed by maturation of the neo orifice. If possible the authors should provide average submucosal lengths.

Author Response

Dear reviewer,
Thank you for your assessment and recommendations.
We have supplemented the information on ureter mobilization and the approximate tunnel length that can be achieved. 
The operations were all performed by C. Kruppa, as indicated except for individual substeps. I have refrained from mentioning the name in the text; if necessary, this should be added.
With best regards
C. Kruppa